# Cellular Responses of Human Lymphatic Endothelial Cells to Carbon Nanomaterials

**DOI:** 10.3390/nano10071374

**Published:** 2020-07-14

**Authors:** Mahoko Sano, Makoto Izumiya, Hisao Haniu, Katsuya Ueda, Kosuke Konishi, Haruka Ishida, Chika Kuroda, Takeshi Uemura, Kaoru Aoki, Yoshikazu Matsuda, Naoto Saito

**Affiliations:** 1Institute for Biomedical Sciences, Interdisciplinary Cluster for Cutting Edge Research, Shinshu University, Nagano 390-8621, Japan; maho.gh10100505@gmail.com (M.S.); 19bs202a@shinshu-u.ac.jp (M.I.); 19hb402j@shinshu-u.ac.jp (K.U.); kominin65@gmail.com (K.K.); 18hb401g@shinshu-u.ac.jp (H.I.); chika.kurokuro@gmail.com (C.K.); tuemura@shinshu-u.ac.jp (T.U.); saitoko@shinshu-u.ac.jp (N.S.); 2Biomedical Engineering Division, Graduate School of Science and Technology, Shinshu University, Nagano 390-8621, Japan; 3Biomedical Engineering Division, Graduate School of Medicine, Science and Technology, Shinshu University, Nagano 390-8621, Japan; 4Division of Gene Research, Research Center for Supports to Advanced Science, Shinshu University, Nagano 390-8621, Japan; 5Physical and Occupational Therapy Division, Graduate School of Medicine, Shinshu University, Nagano 390-8621, Japan; kin29men@shinshu-u.ac.jp; 6Division of Clinical Pharmacology and Pharmaceutics, Nihon Pharmaceutical University, Saitama 362-0806, Japan; yomatsuda@nichiyaku.ac.jp

**Keywords:** carbon nanohorn, carbon black, multi-walled carbon nanotube, lymphatic endothelial cells, cytotoxicity, cellular response

## Abstract

One of the greatest challenges to overcome in the pursuit of the medical application of carbon nanomaterials (CNMs) is safety. Particularly, when considering the use of CNMs in drug delivery systems (DDSs), evaluation of safety at the accumulation site is an essential step. In this study, we evaluated the toxicity of carbon nanohorns (CNHs), which are potential DDSs, using human lymph node endothelial cells that have been reported to accumulate CNMs, as a comparison to fibrous, multi-walled carbon nanotubes (MWCNTs) and particulate carbon black (CB). The effect of different surface characteristics was also evaluated using two types of CNHs (untreated and oxidized). In the fibrous MWCNT, cell growth suppression, as well as expression of inflammatory cytokine genes was observed, as in previous reports. In contrast, no significant toxicity was observed for particulate CB and CNHs, which was different from the report of CB cytotoxicity in vascular endothelial cells. These results show that (1) lymph endothelial cells need to be tested separately from other endothelial cells for safety evaluation of nanomaterials, and (2) the potential of CNHs as DDSs.

## 1. Introduction

Nanomaterials, a 21st century technology, have already been applied industrially in a wide range of fields, and the scope of their application is expected to expand further in the future. In the medical field, a new strategy called nanomedicine, that combines nanotechnology and biomedical science, has been proposed, and it is no exaggeration to say that nanomaterials will play a crucial role in medicine this century [1,2,3]. Carbon nanomaterials (CNMs) are also expected to be applied as a biomaterial in medical applications such as drug delivery systems, regenerative medicine, and biological imaging [4,5,6]. Inevitably, in order to apply CNMs as a biomaterial to medical applications, it is necessary to clarify the effects of CNMs on human health [7,8,9]. Although the safety evaluation of various CNMs has been performed in previous studies using in-vitro and in-vivo experiments, many of these studies focus on safety assessments that only assume occupational inhalation exposure. However, these occupational safety assessments are not always applicable to medical applications of nanomaterials. Furthermore, studies on the use of nanomaterials in imaging systems, drug delivery systems (DDSs), and implant materials have evaluated efficacy, but there are still many uncertainties regarding safety.

Carbon nanohorns (CNHs) are a type of carbon nanotube (CNT) in which a graphene sheet is rolled into a conical shape, has a closed structure at the tip, and forms a horn shape with a diameter of 2–5 nm and length of 40–50 nm [10]. About 2000 of these CNHs are always gathered together, forming a spherical aggregate with a diameter of about 50 to 150 nm. CNHs can be very effective carriers because not only are surface modifications easy, but they can also carry various compounds, including drugs, by oxidization [11,12,13]. In fact, several studies have reported their effectiveness [14,15]. It has also been found that oxidation treatment makes CNHs hydrophilic, reduces aggregation, and enhances biocompatibility [16]. Although there are several reports on the pharmacokinetics of the poorly biodegradable CNHs, details about only relatively large accumulation sites have been revealed due to the poor detection sensitivity of CNHs [17,18,19,20,21].

It is known that macromolecules and waste products, which are present in intercellular spaces and cannot be recovered via blood capillaries, are generally recovered from capillary lymph vessels. Recently, CNTs have also been reported to collect in lymphatic vessels and accumulate in lymph nodes [22,23]. Moreover, metal nanomaterials have been reported to accumulate in reticuloendothelial cells of lymph nodes through nonspecific uptake and remain in the body for more than six months [24]. 

This study evaluated the effects of CNHs, which are expected to accumulate in lymph nodes, on human lymph node-derived endothelial cells. Cytotoxicity, inflammatory response, and uptake into cells of CNHs were examined and compared with those of CNTs, which are known to cause cytotoxicity, and carbon black (CB) as a negative control [4,25]. Moreover, two types of CNHs, untreated and oxidized, were used to evaluate the difference in biological response to the chemical surface properties [26,27].

## 2. Materials and Methods

### 2.1. Carbon Nanomaterials and Their Dispersion

Untreated CNH (unCNH) and oxidized CNH (oxCNH) were purchased from NEC (Tokyo, Japan). The diameters of both CNHs were 50–150 nm, as described by the manufacturer. unCNH and oxCNH have different specific surface areas, unCNH is 400 m^2^/g, and oxCNH is 1300–1400 m^2^/g. CB was provided by Asahi Carbon (Asahi#8; Niigata, Japan) and multi-walled CNT (MWCNT) was provided by Hodogaya Chemical (MWNT-7; Tokyo, Japan). According to the manufacturers’ information, the diameter of CB was 120 nm, and the diameter of MWCNT was 60 nm and the length was 10 µm. These carbon nanomaterials (CNMs) were sterilized in an autoclave at 121 °C for 20 min and dried, then 1 mg/mL was sonicated in 2% fetal bovine serum (Biowest, Nuaillé, France), at which the dispersion state was optimal, in Dulbecco’s phosphate-buffered saline (DPBS) for 60 min using a PR-1 sonicator (Thinky, Tokyo, Japan). Attempts to disperse at concentrations higher than 1 mg/mL could not achieve sufficient dispersion.

To determine the hydrodynamic size of the CNMs, they were measured using a Zetasizer Nano ZS (Malvern Instruments, Malvern, UK). They were diluted to 0.1 mg/mL, and each measurement was conducted in triplicate.

The dispersion state of CNMs was observed using a JEM1400 transmission electron microscope (TEM; JEOL, Tokyo, Japan). CNMs diluted to 0.1 mg/mL were directly dipped in a microgrid. TEM images were captured at 80 kV.

### 2.2. Cell Culture

The human lymphatic endothelial cells (HLECs; lot number 19415) and culture medium (endothelial cell medium) were purchased from ScienCell Research Laboratories (Carlsbad, CA, USA). HLEC culture and experiments used fibronectin-treated dishes, multiplates, and coverslips (BioCoatTM Fibronectin Coating; Corning, Oneonta, NY, USA). For each experiment, the cells within 10 passages, which is the manufacturer’s guaranteed passage number, were seeded at a density of 5 × 10^4^ cells/mL and allowed to adhere for 24 h in a humidified CO_2_ incubator maintained at 37 °C and 5% CO_2_.

### 2.3. Cell Viability

Cell viability was assessed using an AlamarBlue assay (AlamarBlue^®^ cell viability reagent; Remel, Lenexa, KS, USA). Cells were plated in 96-well plates and incubated for 24 or 48 h in culture medium containing 1, 10 or 100 μg/mL CNMs, or in control medium containing only dispersant. After aspiration of the culture medium to exclude the influence of CNMs, 10% AlamarBlue reagent in culture medium was added to each well, where viable cells metabolized the dye for 60 min, resulting in fluorescence detected by excitation/emission at 535/590 nm using a plate reader (AF2200; Eppendorf, Hamburg, Germany). Cell viability was calculated as follows: percent cytotoxicity = 100 × experimental value/control value. Four wells were measured in each group, and the experiment was performed twice.

### 2.4. Inflammatory Response

To evaluate the inflammatory response, mRNA expression of inflammatory cytokines and chemokines was examined. Cells were plated in 24-well plates and incubated with 10 or 100 μg/mL CNMs for 48 h. Total RNA from cells was extracted using a combination of TRIzol^®^ Reagent (Thermo Fisher scientific, Carisbad, CA, USA) and PureLink™ RNA micro kit (Thermo Fisher scientific, Carisbad, CA, USA). Then, 0.35 µg of total RNA was converted to cDNA using a ReverTra Ace qPCR RT Kit (Toyobo, Tokyo, Japan) as per the manufacturer’s instructions. mRNA levels were quantified using the Mastercycler^®^ RealPlex^2^ (Eppendorf) and were performed in 96-well PCR plates, using Thunderbird SYBR qPCR Mix (Toyobo). All specified primers were purchased from TakaraBio (GAPDH: HA067812, IL-1β: HA237369, IL-6: HA209655, MCP-1: HA145737, RANTES: HA258819, IL-8: HA222368, TNF: HA198263; Kyoto, Japan). The following cycling conditions were used: 95 °C for 60 s (polymerase activation), followed by 40 cycles of 95 °C for 15 s and 60 °C for 45 s. Relative mRNA levels were quantified using the formula 2−ΔΔCq, where ΔCq is the difference between the threshold cycle of a target cDNA and an endogenous reference cDNA.

### 2.5. Cellular Uptake

#### 2.5.1. Observation of Cells by Fluorescence Microscopy

Cells cultured in 24-well plates were exposed to 10 μg/mL CNMs for 24 h. After washing the cells three times with DPBS, cells were stained with bisbenzimide H33342 fluorochrome trihydrochloride (H33342, 10 µg/mL; Nacalai Tesque, Kyoto, Japan) and CytoPainter Lysosomal Staining Kit Orange Fluorescence (CytoPainter ab 138895; Abcam, Cambridge, UK) for 1 h. The cells were visualized using an AxioObserverZ1 fluorescence microscope (Carl Zeiss Microscopy GmbH, Jena, Germany) with a 40 × objective lens.

#### 2.5.2. Observation of Cells by TEM

Cells cultured on coverslips were exposed to 10 μg/mL CNMs for 24 h. Cells were fixed with 2.5% glutaraldehyde overnight, washed three times in phosphate buffer (pH 7.4), post-fixed with 1% osmic acid, and embedded in Epon embedding resin (Epok 812; Okenshoji, Tokyo, Japan). Sections were cut to 60 nm thickness, stained with uranyl acetate and lead citrate, and visualized under a JEM1400 TEM (JEOL, Tokyo, Japan) at 80 keV.

#### 2.5.3. Quantification of Intracellular CNMs

We used the method of CNH quantification reported by Yudasaka et al. [28] with some modifications. Cells cultured in 24-well plates were exposed to 50 μg/mL CNMs for 24 or 48 h. To determine the number of viable cells, the fluorescence intensity was measured using the AlamarBlue assay described in Section 2.3, and then washed three times with DPBS. Cells were trypsinized and lysed with 0.5% TritonX-100 in DPBS. Cells in the control group were counted directly before cell lysis. The absorbance of the lysate and the diluted CNMs for the calibration curve were measured at 700 nm using a multi plate reader (VERSA max; Molecular Devices LLC, San Jose, CA, USA). The number of viable cells was calculated from the fluorescence intensity to determine the amount of CNM uptake per cell. 

### 2.6. Statistical Analysis

All data were subjected to Bartlett’s analysis of variance. The cytotoxicity data and the CNM uptake amounts, which were evenly distributed, were subjected to a test of significant difference by the Dunnett method and Tukey-Kramer method, respectively. Cytokine gene expressions that were not evenly distributed were tested for significance by the Steel method. Data are presented as the mean ± S.E. *p*-value < 0.05 was considered statistically significant.

## 3. Results

### 3.1. Characterization of Dispersed CNMs

We have already reported that the dispersion state of CNMs affects some cell lines differently [29,30]; accordingly, we first examined the dispersion state of the CNMs used in the aqueous solution. The sonicated CNMs were measured with a Zetasizer Nano ZS first (Table 1), and their shapes were observed with a TEM (Figure 1). Both of the CNHs were around 150 nm, which was in the range specified by the manufacturers. In the TEM image, most of the CNHs were present as single particles. Although the particle size of CB is described as 120 nm in the catalog, we found it was actually 222.2 ± 111.8 nm when measured in the dispersion. Although aggregation of a small number of CB particles was suspected, it was confirmed from the TEM image that it was well dispersed. The hydrodynamic size of the MWCNTs was 677.5 ± 697.1 nm. Since fibrous nanoparticles such as MWCNTs do not yield accurate fibrous shape information by the dynamic light scattering (DLS) method, it was difficult to determine whether the MWCNT was sufficiently dispersed based on the measured values. Therefore, evaluation by a TEM image is prudent, and it was confirmed that the MWCNT was sufficiently dispersed as shown in Figure 1D. 

### 3.2. Cell Viability

We evaluated the cell viability of HLECs when exposed to each of the CNMs at 10, 50, or 100 µg/mL for 24 or 48 h using the AlamarBlue assay (Figure 2A,B). At 24 h exposure, all groups except 100 µg/mL oxCNH had 90% cell viability or higher. The viability of cells treated with CNH at 100 µg/mL was also 89.8% ± 3.5%, which was not a significant difference compared with the control. At 48 h exposure, 68.2% ± 1.2% and 64.2% ± 7.2% the cells exposed to 50 and 100 µg/mL MWCNT, respectively, were viable, showing a significant decrease compared to the control group. The other groups had 90% or more cell viability even at the highest concentration of CNM, and no cytotoxicity in HLECs was observed.

### 3.3. Inflammatory Responses

The expression of six inflammatory cytokines (IL-1β, IL-6, MCP-1, RANTES, IL-8 and TNF) was measured after exposure to each CNM at 10 and 100 µg/mL. Four inflammatory cytokines showed a significant change in gene expression compared to the control group (Figure 3). Exposure to 100 µg/mL MWCNT significantly increased expression of IL-6, RANTES, IL-8, and TNF compared to the control group. In particular, gene expression of IL-6 was increased approximately 72-fold and IL-8 expression was increased approximately 14-fold compared to the control group. Moreover, IL-6 and RANTES also showed a significant increase in gene expression in cells exposed to 10 µg/mL MWCNT. The unCNH and oxCNH groups did not show any significant changes compared with control or concentration-dependent trends.

### 3.4. Cellular Uptake

#### 3.4.1. Observation of Cells by Fluorescence Microscopy

We observed the condition of HLECs exposed to each CNM at 10 or 100 µg/mL by fluorescent staining of lysosomes and nuclei. Phase contrast and fluorescence images were merged such that the localization of lysosomes and nuclei in cells could be determined, and a bright-field image showing the localization of CNMs was also acquired (Figure 4). In the images of unCNH-, oxCNH-, and CB-treated cells, the highly refractive regions indicated by arrows in the merged images corresponded with black regions in the bright-field images, indicating that they were CNMs. In the phase contrast image of the MWCNT, it can be confirmed that the aggregated MWCNT appears black in addition to the regions that appear white. Most CNMs co-localized with red-stained lysosomes, but some unCNH, oxCNH, and CB were present in the cytoplasm and which were presumed not to have merged yet with the lysosomes. Some MWCNTs appeared to protrude from the lysosome in a black aggregated state. In addition, for all types of CNMs, some cells took up a very large amount of CNM, while other cells took up only a small amount of CNM, suggesting that the uptake by individual cells varied widely. unCNH, oxCNH, and CB also appeared to be taken up more readily when concentration was increased 10-fold. Contrastingly, even at 10 μg/mL, a significant amount of MWCNT was taken up into cells, and uptake did not increase further with a 10-fold increase in concentration. 

#### 3.4.2. Observation of Cells by TEM

Figure 5 shows a TEM image of HLECs exposed to CNMs at a concentration of 10 µg/mL. As in the observation under a fluorescence microscope, it was confirmed that all types of CNMs were internalized to some extent by the cells. Compared to oxCNH, unCNH remained in the shape of sea urchin spikes, whereas oxCNH was highly particulate with a collapsed spine structure. Many CB particles seemed to be aggregated into several pieces. However, HLECs, which seemed to have accumulated aggregate CB in lysosomes, had holes that were traces of the removal of aggregated CB due to thin sectioning (data not shown). The MWCNT was visible as a cross section with a long diameter, and many ‘scratches’ and ‘holes’ were observed that were thought to be MWCNTs.

#### 3.4.3. Quantification of Intracellular CNMs

Figure 6 shows the uptake of CNMs by HLECs exposed to 50 μg/mL CNM for 24 or 48 h. Regardless of time, MWCNT had the highest uptake. It was significantly higher than that of other groups, and in particular was 5 to 6 times greater than uptake of both CNHs. CB had the next highest intracellular uptake, which was also significantly greater than that of the two CNH groups. The uptake of oxCNH was slightly higher than that of unCNH, but this difference was not significant. The increase in cellular uptake as a function of time was 1.9-fold for MWCNT and oxCNH, whereas it was about 1.8-fold for unCNH, and less than 1.5-fold for CB. In addition, the uptake rate (% CNM taken up) was highest with CB at 19.8 ± 1.2%, followed by MWCNT, oxCNH, and unCNH in the at 16.4 ± 0.8%, 5.6 ± 1.0%, and 4.6 ± 0.9%, respectively.

## 4. Discussion

It is known that CNMs containing CNHs are non-biodegradable and are retained in vivo [31]. Therefore, when CNMs are used in DDSs, it is thought that the fraction that is not used in the target tissue has two possible fates. The CNHs may be recognized as foreign substances and taken up by macrophages, or they may be collected into the lymphatic system like waste products and lipids. Although there have been studies into the effects of CNM uptake on macrophages and the accumulation of CNM in lymph nodes [31], there have been no reports up to now on the effects of CNM uptake on cells constituting the lymph nodes. The absence of any such reports does not mean that CNMs do not adversely affect the living body. Some studies have reported the disease risk of tattoo inks that accumulate in lymph nodes [32,33], and the main component of tattoo ink is CB. Therefore, we evaluated the cytotoxicity of two types of CNHs in HLECs by comparing them with CB and with MWCNT, which has been reported to have cytotoxicity and carcinogenicity.

As for the effect of CNMs on proliferation of HLECs, only MWCNT showed toxicity to proliferation. We have already reported the effects of MWCNT and CB on cell proliferation in several cell types [34]. Although the detailed conditions such as the type of CB and MWCNT, the dispersant, and the dispersion state differed, HLECs showed cytotoxicity in response to MWCNTs but not CB, which was similar to the response of the various cell types that internalized CNMs in our previous study. On the other hand, there is a lack of consensus in cytotoxicity assessments of CNMs in various cell types [35]. Flahaut et al., used human umbilical vein endothelial cells (HUVECs) to evaluate the cytotoxicity of MWCNT and reported that there was no cytotoxicity [36]. However, the concentrations of MWCNT they evaluated were below 1 μg/mL, which is not comparable to this study. Zhao et al., evaluated the cytotoxicity of MWCNT in HUVECs and reported that 64 μg/mL of thin MWCNT was cytotoxic [37]. Although they reported that MWCNT as thick as that used in the present study was not cytotoxic, they used a type of MWCNT that was considerably shorter than what we used, and it is difficult to directly compare our results. In contrast, Yamawaki et al. reported that two types of particulate CNM, CB (Φ248.2 ± 161.4 nm) and fullerene (Φ7.1 ± 2.4 nm), were evaluated in HUVECs for their capacity to promote cell proliferation, and both inhibited cell growth [38,39]. Cells exposed to 100 μg/mL hydroxyl fullerene for 24 h showed 64.2% cell growth inhibition by ubiquitin-autophagic cell death, and 46.7% cell growth inhibition at the same concentration of CB exposure but without induction of ubiquitin-autophagic cell death. These results from studies of CNMs in HUVECs differ from ours. We think this difference is due to cell type differences. Lymphatic vessels are formed embryologically by the sprouting of endothelial cells that form venous blood vessels [40]. Therefore, although the two cell types have some common features, differences are in their responses to CNMs are apparent [41]. Autophagic vacuoles, which are thought to have effected cell death in response to fullerenes, were also observed in the control group in HUVECs, but they were not present in the lymph node-derived endothelial cells we used as shown in Figure 4A. In addition, PLVAP, which is one of the markers that distinguish between venous endothelial cells and lymphatic endothelial cells, is a key protein in the formation of caveolae diaphragms, fenestrae and trans-endothelial channels for vascular permeability, and is not expressed in lymphatic endothelial cells [42,43,44]. At least, endocytosis involving this molecule would be unlikely to occur in HLECs. Caveolae are involved in endocytosis and are reported not to fuse with lysosomes, unlike other endocytoses [45,46]. Based on the fact that most CB and CNHs were taken up by lysosomes in HLECs, it is apparent that there are differences in the mechanism of uptake between HUVECs and HLECs. Therefore, their proliferative responses may differ.

Differences in cell types clearly influence the inflammatory response to CNMs. Yamawaki et al. also measured the secretion of MCP-1 protein to assess the inflammatory response of HUVECs to CB and reported a more than 6-fold increase [39]. We measured the expression of six inflammation-related genes, including MCP-1, in HLECs, but CB exposure only increased IL-8. Since even CNTs did not affect the expression of MCP-1 in HLECs, the increase in secretion of MCP-1 protein by CB exposure in HUVECs may be mediated by cytokines specifically expressed in HUVECs. Alternatively, it is possible that regulation of MCP-1 by CNMs occurs at a translational or post-translational level and is not reflected in mRNA expression. Contrastingly, in HUVECs, which showed cytotoxicity when exposed to MWCNT [37], IL-6 secretion was increased, as was IL-6 gene expression in HLECs. Cytotoxicity due to nanocarbon has been reported to promote secretion of cytokines [35]. However, the expression of IL-8 is increased not only in response to MWCNT but also in response to CB in which cytotoxicity is not observed. We reported that in differentiated THP-1 cells, which internalized CNMs, MWCNT and CB did not impair IL-8 protein secretion, while IL-8 secretion was increased in undifferentiated THP-1 cells without cellular uptake of the CNMs [34]. Furthermore, IL-8 is a cytokine that is constantly stored in endothelial cell-specific organelles called Weibel-Palade bodies [47,48]. It is speculated that MWCNT and CB are recognized as inflammatory triggers on the cell membrane of endothelial cells [49]. However, CNHs, which are almost the same size as CB particles, are not recognized as an inflammatory trigger. This recognition mechanism seems to distinguish CB from CNHs.

Is the internalization mechanism of different CNMs in HLECs the same? Of the CNMs used in this study, the uptake weight of MWCNT was the highest at both time points. We previously compared the uptake of well-dispersed MWCNTs and CNHs by macrophage-like Raw264 cells, and reported that CNH was taken up but MWCNT was hardly taken up [30]. These results indicate that the cellular uptake mechanism of fibrous and spherical CNMs varies depending on the cell type. Comparing the rates of increase in CNM uptake between 24 and 48 h, MWCNTs and CNHs are comparable, but CB has a low rate of increase. Considering that suppression of cell growth influences the increase in uptake rate of MWCNT, the uptake rate of CNHs is actually highest. We have reported that intracellular uptake of MWCNT in non-phagocytic cell lines may be due to three types of endocytosis as well as uptake by alternative pathways [50], and Liu et al., recently published a review of intracellular uptake of nanoparticles [51]. However, the molecular mechanism by which cells first recognize nanomaterials remains unclear. Although there are reports that scavenger receptors MARCO and SCARB1 are involved in CNM uptake by cells expressing those receptors [52,53], the Human Protein Atlas states their expression site in cells is intracellular membrane, not the cytoplasmic membrane [54]. Although the CB and CNHs used in this study have similar particle sizes and densities, it is very interesting that the amount and speed of cellular uptake differ. It is possible that the composition of the protein corona formed on the particle surface may be different [55,56], or there may be a mechanism that recognizes the difference in the shape of the particle surface. This study uses two types of CNHs, untreated and oxidized, and their protein coronas are expected to be different, but further studies are needed.

There was no significant difference in biological responses of HLECs to CNHs with different surface treatments. Holder et al., evaluated the effect of soot with a particle size of approximately 280 nm, with and without oxidation, on bronchial epithelial cells [57]. They report that oxidized soot was more cytotoxic than untreated, but did not differ in its induction of IL-8 secretion. In contrast, Eldridge et al. evaluated toxicity by exposing the microvascular endothelial cells with short and thin MWCNT, with or without oxidation treatment and with or without phospholipid-polyethylene glycol (PL-PEG) coating [58]. They reported that coated MWCNT was cytotoxic, with greater toxicity from untreated MWCNT, whereas uncoated MWCNT was not cytotoxic with or without oxidation treatment. We reported that bronchial epithelial cells are extremely sensitive to CNMs and that cytotoxicity is associated with the amount of CNM internalized [34,59]. Although there was no significant difference in the intracellular uptake of the two CNHs in this study, oxCNH was more abundant at both time points. In HLECs, the effect of oxidative treatment may be more apparent when the amount of unCNH taken up is equal to the amount of CB that causes cytotoxicity. We believe that the difference in the shape of the CNMs caused the difference in toxicity since CNHs and CB in the same particulate form did not show cytotoxicity under the concentration conditions generally performed in vitro. Conversely, the difference in the chemical properties due to oxidation treatment may make a difference in PL-PEG-coated CNMs. The two types of coated MWCNTs induce cytotoxicity with almost no uptake into cells, indicating a mechanism completely different from that of intracellular cytotoxicity. It is speculated that the direct effect of oxidative treatment of particulate CNMs on the biological response is rather limited. 

Assuming the use of CNMs in vivo, we evaluated the toxicity of CNM in HLECs, where its accumulation is a concern. Cytotoxicity was observed in fibrous MWCNT, but severe cytotoxicity was not observed in spherical CB and CNHs. In addition, neither oxidized nor unoxidized CNHs accumulate much in cells, suggesting it may be very suitable for medical applications. However, recent studies have revealed that there are at least six types of lymphatic endothelial cells, and it is not clear which of these lymphatic endothelial cells was used [60]. More detailed studies are needed in the future, including the effects of long-term exposure. 

## 5. Conclusions

We performed risk assessment in HLECs as part of the safety assessment of CNMs, which are expected to have medical applications. Even if a large amount of particulate CNM was accumulated in HLECs, almost no cytotoxicity was observed, and this is a promising result for future medical applications.

## Figures and Tables

**Figure 1 nanomaterials-10-01374-f001:**
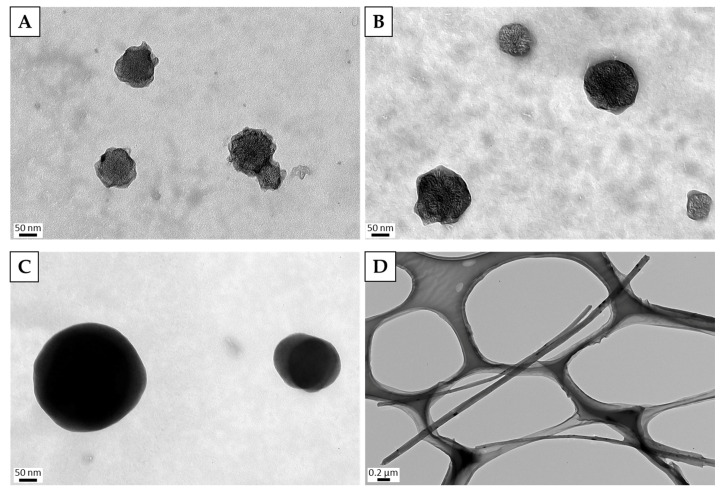
TEM images of carbon nanomaterials dispersed by sonication for 1 h. (**A**) unCNHs, (**B**) oxCNHs, (**C**) CB, (**D**) MWCNTs. TEM, transmission electron microscope; unCNH, untreated carbon nanohorns; oxCNHs, oxidized carbon nanohorns; CB, carbon black; MWCNTs, multi-walled carbon nanotubes.

**Figure 2 nanomaterials-10-01374-f002:**
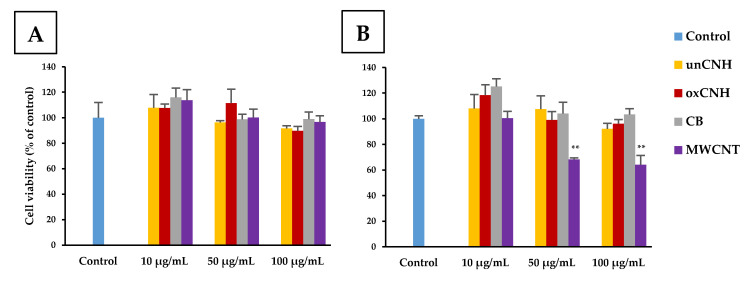
Cell viability of HLEC exposed to varieties of carbon nanomaterials. (**A**) The cell viability was measured for 24 h. (**B**) The cell viability was measured for 48 h. Data are expressed as the mean ± S.E. (*n* = 4). ** *p* < 0.01. HLEC, human lymphatic endothelial cell; unCNH, untreated carbon nanohorn; oxCNH, oxidized carbon nanohorn; CB, carbon black; MWCNT, multi-walled carbon nanotube.

**Figure 3 nanomaterials-10-01374-f003:**
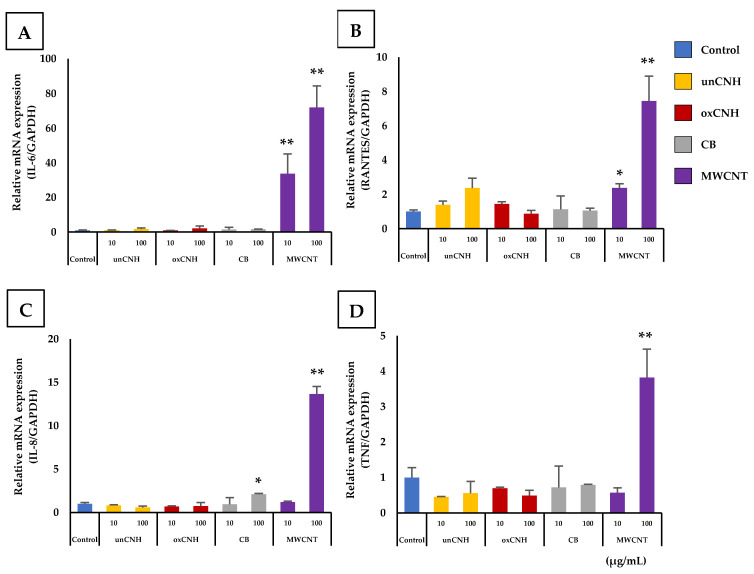
Gene expression of inflammation cytokines on HLEC exposed to varieties of carbon nanomaterials. (**A**) IL-6, (**B**) RANTES, (**C**) IL-8, (**D**) TNF. Data are expressed as mean ± S.E. (*n* = 3). * *p* < 0.05; ** *p* < 0.01. HLEC, human lymphatic endothelial cell; unCNH, untreated carbon nanohorn; oxCNH, oxidized carbon nanohorn; CB, carbon black; MWCNT, multi-walled carbon nanotube.

**Figure 4 nanomaterials-10-01374-f004:**
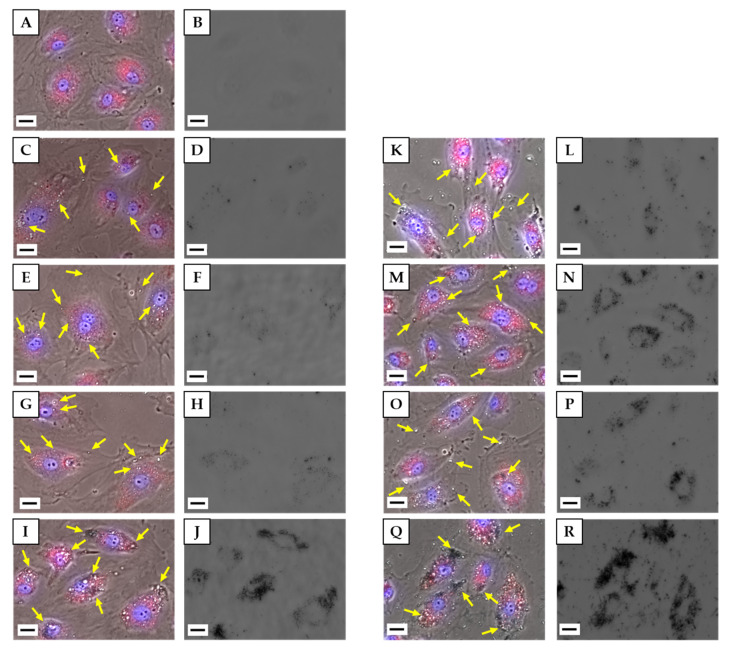
Live HLEC images after incubation with H33342 for nuclear staining and CytoPainter LysosomalStaining Kit for lysosomes. (**A**,**B**) Control, (**C**,**D**,**K**,**L**) unCNH, (**E**,**F**,**M**,**N**) oxCNH, (**G**,**H**,**O**,**P**) CB, (**I**,**J**,**Q**,**R**) MWCNT. (**A**,**C**,**E**,**G**,**I**,**K**,**M**,**O**,**Q**) Image obtained by merging the phase contrast image and the fluorescence images. (**B**,**D**,**F**,**H**,**J**,**L**,**N**,**P**,**R**) Bright field image. (**C**–**J**) 10 µg/mL. (**K**–**R**) 100 µg/mL. Scale bars correspond to 20 μm. Yellow arrows indicate CNMs. HLEC, human lymphatic endothelial cell; H33342, bisbenzimide H33342 fluorochrome trihydrochloride; unCNH, untreated carbon nanohorn; oxCNH, oxidized carbon nanohorn; CB, carbon black; MWCNT, multi-walled carbon nanotube; CNMs, carbon nanomaterials.

**Figure 5 nanomaterials-10-01374-f005:**
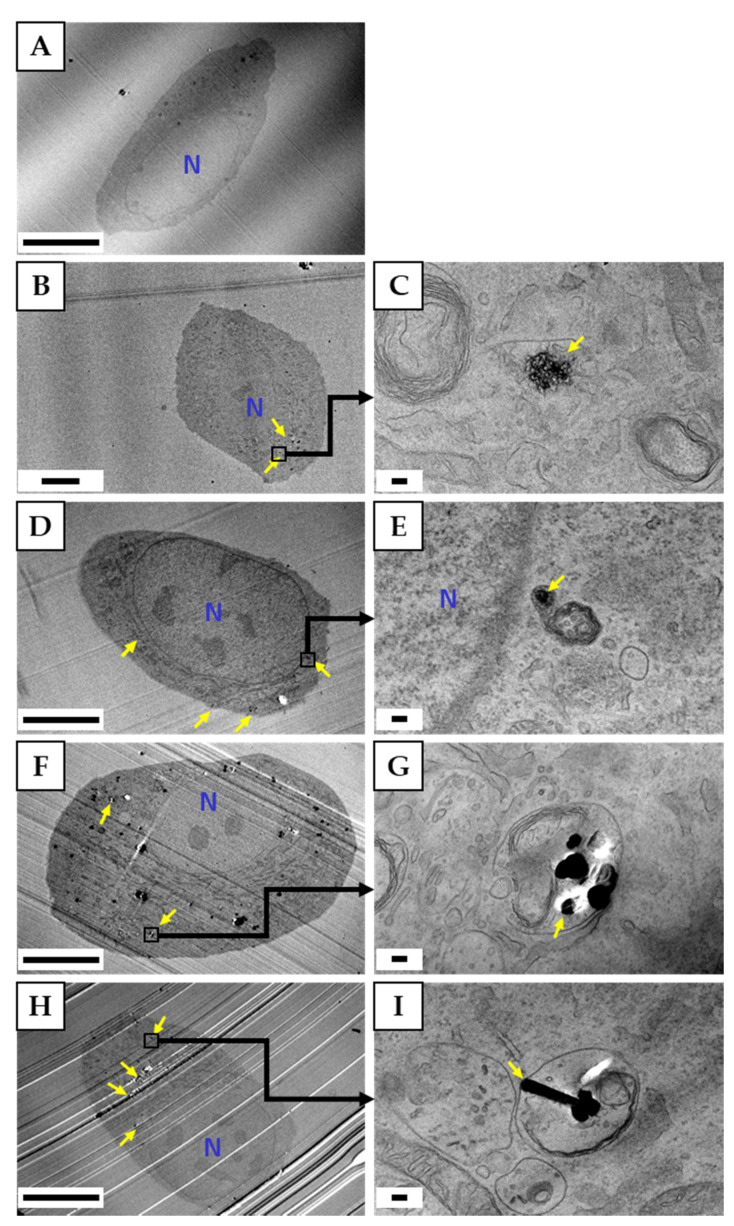
TEM images of HLECs exposed to CNMs at 10 μg/mL. (**A**) Control, (**B**,**C**) unCNH, (**D**,**E**) oxCNH, (**F**,**G**) CB, (**H**,**I**) MWCNT. (**A**,**B**,**D**,**F**,**H**) Low magnification. Scale bars correspond to 10 μm. (**C**,**E**,**G**,**I**) High magnification. Scale bars correspond to 100 nm. Yellow arrows indicate CNMs. TEM, transmission electron microscope; HLEC, human lymphatic endothelial cell; CNMs, carbon nanomaterials; unCNH, untreated carbon nanohorn; oxCNH, oxidized carbon nanohorn; CB, carbon black; MWCNT, multi-walled carbon nanotube; N, nucleus.

**Figure 6 nanomaterials-10-01374-f006:**
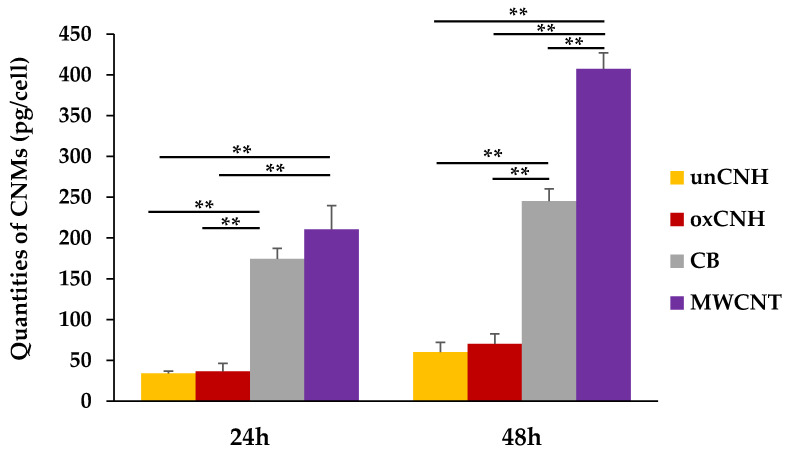
Uptake of each CNM per cell. HLECs were exposed to 50 μg/mL CNMs for 24 h and 48 h. Mean ± S.E. (*n* = 4), ** *p* < 0.01. CNM, carbon nanomaterial; HLEC, human lymphatic endothelial cell; unCNH, untreated carbon nanohorn; oxCNH, oxidized carbon nanohorn; CB, carbon black; MWCNT, multi-walled carbon nanotube.

**Table 1 nanomaterials-10-01374-t001:** The hydrodynamic size of CNMs measured by Zetasizer.

CNMs	Average Particle Size (nm)
unCNH	155.5 ± 61.6
oxCNH	150.7 ± 55.6
CB	222.2 ± 111.8
MWCNT	677.5 ± 697.1

CNMs, carbon nanomaterials; unCNH, untreated carbon nanohorn; oxCNH, oxidized carbon nanohorn; CB, carbon black; MWCNT, multi-walled carbon nanotube.

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
