# Peer review of "Cellular Responses of Human Lymphatic Endothelial Cells to Carbon Nanomaterials"

_nanomaterials, 2020, doi:10.3390/nano10071374_

Round 1

Reviewer 1 Report

The paper of Sano et al., entitled ”Cellular responses of human lymphatic endothelial cells to carbon nanomaterials” is a very interesting approach regarding carbon nano materials that have an important potential as drug delivery systems. The paper is appropriately designed and highlights the up-take of CB, CHN and MWCNT and their effects on HLEC cells. 

Even if it very well written, the paper presents some leaks in writing. So, the authors have to specify which is the specific culture medium used for HLEC cells.

They pretend that all carbon nano materials are not toxic for these cells that is not sustained by the results.

Author Response

I made the response to the comments an attachment.

Reviewer 2 Report

Comments to the Author

General consideration

The authors present data about the evaluation of carbon based-nanomaterials toxicity on human lymph node endothelial cells. The results demonstrated that fibrous multi-walled carbon nanotubes show, at higher concentration, cell proliferation suppression, as well as expression genes related to inflammation. No significant toxicity was observed for particulate carbon black and carbon nanohorns. However, the results reported in the manuscript lack of novelty and originality. Furthermore, it is not clear what is the difference between the two species of carbon nanohorns and how the chemical surface properties influence the biological response.

Below there are some suggestions:

Major revision

-             The authors reported results obtained only on lymph endothelial cells. A comparison between two or three lymphatic cell lines is necessary in order to define the toxicity of the materials at the lymphatic level.

-             Introduction: In the introduction section, the authors should better emphasize the chemical surface properties between the two species of carbon nanohorns and how this can influence the biological response.

Author Response

(The authors gave the same response as above.)

Reviewer 3 Report

Dear Authors,

I have to admit that I expected something more spectacular, and here first sentence and first sin:

“ The medical application of carbon nanomaterials (CNMs) has been anticipated for more than 15 years”. – you mean 16? 25? 50?

The 2nd sin:

“However, the pharmacokinetics of CNHs, which are poorly biodegradable, are not well understood [16,17].”

There is a lot of works concerning CNM cytotoxicity (including CNH), and you referred only two

What do you meant “…… carbon black (CB) as a negative control.”

Before I will read further, I would like to know something about the tested materials.

the chapter 2.1 is very poor and phys-chem characteristic is limited to, not very well performed 2 experiments: TEM (by the way in Fig 1D - these are not CNTs, most probably this is the net), and Zeta-measurements with extremely high concentrations of materials coated with proteins, why?

Please rebuild the chapter “2.1. Carbon Nanomaterials and their Dispersion”

  • What does it mean oxidized? – conditions
  • CB? – and the only information is diameter, why?
  • When your CNM were “sonicated in 2% fetal bovine serum” then “hydrodynamic size” was measured for complex, not for CNM alone
  • please look closer to the concentration “They were diluted to 1 mg/mL ……” seems to be not real

This could be a good work, however I would expect the answer "Why?", why is it non-toxic? or Why is it toxic? What is the mechanism of the process? Thus, it is crucial to combine yours in-vitro results with phys-chem characteristic. Otherwise you have 3 unknown samples 2 of them are non-toxic the 3rd is little cytotoxic there is a lot of such works in the literature.

good luck

Author Response

(The authors gave the same response as above.)

Round 2

Reviewer 2 Report

The authors add a review of the literature which reports the results obtained for the same type of materials on deferred cells. Otherwise, they must enter at least one other cell line to be able to say that the materials exhibit different behavior.

Author Response

The authors add a review of the literature which reports the results obtained for the same type of materials on deferred cells. Otherwise, they must enter at least one other cell line to be able to say that the materials exhibit different behavior.

Thank you very much for this suggestion. We have cited the following current review containing other cell evaluations of CNMs, including the CNMs that we used in this study. This review has been mentioned in the Discussion section as an evaluation of CNMs in other cells.

  • Yuan, X.; Zhang, X.; Sun, L.; Wei, Y.; Wei, X. Cellular Toxicity and Immunological Effects of Carbon-based Nanomaterials. Part Fibre Toxicol 2019, 16, 18.

Reviewer 3 Report

Dear Authors,

Thank you for considering my remarks and answering my questions. Nowadays problem of toxicity of CNM is a high topic. On the one side there is the problem of "the shape" but on the other is the surface chemistry, thus please discuss your results more detail with the literature as e.g.: DOI: 10.3390/MA13092060, DOI: 10.2174/1389557519666191029162150, DOI: 10.1007/s12010-015-1607-1

Best wishes,

Author Response

Thank you for considering my remarks and answering my questions. Nowadays problem of toxicity of CNM is a high topic. On the one side there is the problem of "the shape" but on the other is the surface chemistry, thus please discuss your results more detail with the literature as e.g.: DOI: 10.3390/MA13092060, DOI: 10.2174/1389557519666191029162150, DOI: 10.1007/s12010-015-1607-1

Thank you for raising this important point. Indeed, we evaluated both unCNHs and oxCNHs in this study because we also wanted to consider the surface chemistry pointed out by the Reviewer. In the Introduction section, we have included the two papers that the Reviewer indicated as references for the basis of using two types of CNHs.

  • Werengowska-Ciećwierz, K.; WiÅ›niewski, M.; Terzyk, A.P.; Roszek, K.; Czarnecka, J.; Bolibok, P.; Rychlicki, G. Conscious Changes of Carbon Nanotubes Cytotoxicity by Manipulation with Selected Nanofactors. Appl Biochem Biotechnol 2015, 176, 730-741.
  • Czarnecka, J.; WiÅ›niewski, M.; Forbot, N.; Bolibok, P.; Terzyk, A.P.; Roszek, K. Cytotoxic or Not? Disclosing the Toxic Nature of Carbonaceous Nanomaterials through Nano-Bio Interactions. Materials (Basel) 2020, 13, 2060.